# Genome-Wide and 16S rRNA Sequencing-Based Analysis on the Health Effects of *Lacticaseibacillus paracasei* XLK401 on Chicks

**DOI:** 10.3390/microorganisms11092140

**Published:** 2023-08-23

**Authors:** Xin Kang, Xin-Dong Li, Huan-Yu Zhou, Feng Wang, Lian-Bing Lin

**Affiliations:** 1Faculty of Life Science and Technology, Kunming University of Science and Technology, Kunming 650500, China; kx1012294117@icloud.com (X.K.); pxyhelxd@icloud.com (X.-D.L.); 13224022625@163.com (H.-Y.Z.);; 2Engineering Research Center for Replacement Technology, Feed Antibiotics of Yunnan College, Kunming 650500, China

**Keywords:** growth promoter, probiotic potential, genome-wide, 16S, *Lacticaseibacillus paracasei*

## Abstract

*Lacticaseibacillus paracasei*, serves as a growth promoter used in the poultry industry, contributeing to broiler development. However, practical studies are needed to determine the probiotic potential and growth-promoting effects of specific *L. paracasei* strains. This study aims to determine whether *L. paracasei* XLK401 influences broiler chicken growth and the mechanisms involved. Notably, we identified several bile salt and acid tolerance-related genes (*Asp23*, *atpD*, *atpA*, *atpH*, and *atpF*) in *L. paracasei* XLK401. This bacterium demonstrates robust probiotic properties under acidic conditions (pH 2.0) and 0.3% bile salt conditions. It also contains a variety of antioxidant-related genes (*trxA*, *trxB*, and *tpx*), carbohydrate-related genes, gene-encoding glycosidases (e.g., *GH* and *GT*), and three clusters of genes associated with antimicrobial compounds. Supplementation with *L. paracasei* XLK401 significantly increased the body weight of the chicks. In addition, it significantly increased hepatic antioxidant enzyme activities (GSH-Px, SOD, and T-AOC) while significantly decreasing the levels of oxidative damage factors and inflammatory factors (MDA and IL-6), resulting in improved chick health. Improvements in body weight and health status were associated with significant increases in α-amylase activity and the remodeling of the host gut microbiota by *L. paracasei* XLK401. Among them, actinobacteria abundance in chicken intestines after feeding them *L. paracasei* XLK401 was significantly decreased, *Bifidobacterium sp.* abundance was also significantly decreased, and *Subdoligranulum* sp. abundance was significantly increased. This suggests that *L. paracasei* XLK401 can regulate the abundance of certain bacteria without changing the overall microbial structure. In addition, in the correlation analysis, *Subdoligranulums* sp. were positively correlated with SOD and negatively correlated with IL-1β and MDA. Overall, our study demonstrates that *L. paracasei* XLK401 effectively promotes healthy chick growth. This is made possible by the modulation of gut microbe abundance and the underlying probiotic effect of *L. paracasei* XLK401. Based on these findings, we postulate *L. paracasei* XLK401 as a potential efficient growth promoter in broiler farming.

## 1. Introduction

Since 2006, the European Union has prohibited the use of antibiotics in animal feed to boost growth [1]. This ban has resulted in a significant rise in disease outbreaks in broiler farming [2]. Therefore, there is an urgent need to find effective antibiotic alternatives for use in the poultry industry. It was found that fiber-degrading enzymes, prebiotics, and probiotics can serve as alternatives to antibiotics, with probiotics being more advantageous due to their low production cost and wide range of applications in different species of host animals [3].

In previous studies, probiotics have exhibited a variety of effects, including bacterial growth inhibition, immune response enhancement, and growth performance promotion [4]. For instance, dietary supplementation with *Bacillus subtilis* reduces the proliferation of *Clostridium* perfringens in the gastrointestinal tract of broiler chickens [5]. Similarly, peptidoglycan derived from *Lacticaseibacillus rhamnosus* has been able to induce antimicrobial peptide defensins while avoiding inflammatory responses in broilers [6]. Additionally, as a probiotic feed additive, *Enterococcus faecalis* L3 can increase the average body weight of chickens [7]. However, it should be noted that the physiological effects of probiotics can vary depending on the specific probiotic strain, and therefore certain effects observed for one strain cannot be generalized to other bacteria of the same species or genus. In particular, genetic diversity leads to the presence or absence of specific functions in different strains, necessitating the analysis and evaluation of the relevant features of their probiotic functions at the genetic level [8].

*L. paracasei*, as a potential probiotic, has been reported to exhibit antimicrobial activity, immune system stimulation effects, anti-inflammatory activity, antioxidant activity, and modulatory effects on the intestinal microbiota [9]. For example, *L. paracasei* CNCM I-1518 was found to have an elevated survival rate in the gastrointestinal tracts of mice and humans, an important marker of a potential probiotic strain [10]. The in vivo gastroprotective effect of *L. paracasei* CIDCA 8339 not only involves a direct interaction with gastric mucosa, but also the in situ production of other metabolites that can modulate inflammatory response [11]. The *L. paracasei* CT12 cell-free supernatant isolated from Mexican yogurt pellets has antibacterial and antioxidant effects [12]. *L. paracasei* can also be useful in inflammatory bowel disease treatment through immunomodulation and increases in the intestinal microbiota [13]. However, while these studies suggest that certain *L. paracasei* strains are safe for human use and provide specific health benefits to the host, these positive effects cannot be extended to other strains in the absence of experimental data.

Therefore, this study leverages whole genome sequencing and animal testing to thoroughly and systematically assess the probiotic benefits of *L. paracasei* XLK401. First, the whole genome of *L. paracasei* XLK401 is sequenced to assess the strain’s potential properties at the genetic level. Subsequently, feeding experiments are conducted to evaluate the effects of *L. paracasei* XLK401 on growth performance, antioxidant capacity, immune factors, and the gut microbiota in Nandan-Yao chickens. This study may provide a theoretical basis for *L. paracasei* XLK401 to promote the growth of broiler chickens.

## 2. Materials and Methods

### 2.1. Bacterial Strains

The probiotic used was *L. paracasei* XLK401 (GenBank No. CP098411.1), which was stored in the Engineering Research Center for Feed Antibiotic Replacement Technology of Kunming University of Science and Technology.

### 2.2. Whole Genome Sequencing and Genome Annotation

Prior to bacterial DNA extraction, the culture was centrifuged at 8000× *g* for 40 min to obtain a bacterial precipitate. Subsequently, the bacterial precipitate was washed three times with sterile phosphate buffer. Total genomic DNA of *L. paracasei* XLK401 was extracted using the sodium dodecyl sulfate (SDS) method in conjunction with a purification column. The whole genomic DNA was sequenced using an ONT PromethION sequencer (Oxford Nanopore Technologies, Oxford, UK). For filtering purposes, low-quality and short-length reads were discarded from the original reads. The reads were assembled using Unicycler V0.4.9 (version V0.4.9, Beijing, China) software [14]; the coding genes of the assembled genome were predicted by Prokka (Version 1.12, Beijing, China) [15] software. RepeatMasker V4.1.0 software was used to predict repetitive sequences in the *L. paracasei* XLK401 genome. PseudoFinder software (Version 1.1.0) was used for the prediction of *L. paracasei* XLK401 pseudogenes. Clusters of regularly interspaced palindromic repeats (CRISPRs) on the *L. paracasei* XLK401 chromosome were predicted using MinCED V0.4.2 software. Genomic islands in the *L. paracasei* XLK401 genome were identified by IslandViewer 4 (http://www.pathogenomics.sfu.ca/islandviewer/. Accessed 1 December 2022). Phages in the *L. paracasei* XLK401 genome were predicted using PhiSpy (https://github.com/linsalrob/PhiSpy. Accessed 1 December 2022).

Predicted genes were aligned to various functional databases using BLAST. Alignments were annotated with the highest scores (default identity ≥40%, coverage ≥40%) based on the BLAST results for each sequence. Predicted coding genes were annotated against the GO (Gene Ontology), KEGG (Kyoto Encyclopedia of Genes and Genomes), COG (Cluster of Protein Orthologous Groups), NR (Non-Redundant Protein Database), TCDB (Transporter Proteins Classification Database), and CAZy (Carbohydrate-Activated EnZymes Database) functional databases, as well as the Pathogen-Host Interaction Database. Based on the results of the abovementioned sequencing and bioinformatics analyses, circular mapping of the *L. paracasei* XLK401 genome was performed. Finally, the secondary metabolism gene cluster was analyzed using the secondary metabolite database (antiSMASH) (v5.2.0), while the bacteriocin synthesis gene cluster of the strain was analyzed using BAGEL4 [16].

### 2.3. In Vitro Tolerance Test of L. paracasei XLK401

Intestinal and gastric fluid tolerance determinations were performed by adopting the method described by Li X. Y. et al. [17]. Briefly, for bile salt tolerance tests, different concentrations (0.3, 0.6, and 0.9% (*w*/*v*) of bile salts (Sangon Biotech, Shanghai, China) were added to MRS broth medium; for acid-resistance tests, MRS broth medium was adjusted to different pH values (pH 2.0, 3.0, 4.0, and 6.8) using 2 mol/mL HCl. Subsequently, *L. paracasei* XLK401 cultures (10^7^ CFU/mL) were added to the MRS broth medium and cultured at 37 °C for 4 and 5 h, respectively. After incubation, 100 μL of each bacterial suspension was separately coated on MRS solid plates by the serial dilution method and inverted incubation at 37 °C for 24 h. The viable cell counts of 30 ~ 300 colonies were determined, and tolerance was determined by calculating the ratio (%) of viable cells compared to the control without additives, that is, the survival rates.

### 2.4. Birds, Diet, and Experimental Design

All experiments were performed in accordance with the National Animal Care and Use Guidelines approved by the National Bureau of Animal Research approved by the Ethical Committee for Animal Experimentation of Kunming University of Science and Technology, No. SYXK K2018-0008. Each treatment group consisted of 60 1-day-old Nandan scallop chickens. The dietary treatments included (1) a blank control group (NC; basal diet) (10/cage) and (2) treatment group (FGL; basal diet + *L. paracasei* XLK401 (1 × 10^7^ CFU/g feed) (10/cage). The basal diet was composed of soybean meal, corn, soybean oil, vitamins, and mineral premix and prepared according to the National Research Council (NRC) standard [18] (Table 1). Feeding and sampling were conducted for a total of 22 days. The feed used was purchased from Guang Hong Feed Co., Ltd., in Kunming, China, to meet the full feed requirements for 0–21-day-old chicks. These diets were provided to the chickens as crumbled pellets.

### 2.5. Growth Performance Evaluation

All birds and feeds were weighed at 0, 7, 14, and 21 d. The feed consumption and body weight were recorded on a pen basis for each treatment group. Final body weight (BW) and feed conversion ratio (FCR = feed consumption/weight gain) were calculated.

### 2.6. Sample Collection

After 12 h of fasting, the chickens were sacrificed and the blood, liver, duodenal segments, and cecum were harvested. For the analysis of immune factor concentrations, blood samples (3 mL per bird) were centrifuged at 4 °C and, 1500× *g* for 15 min to obtain serum (0.5 mL per bird), which was then quickly frozen in liquid nitrogen and stored at −80 °C for the subsequent analysis. Liver samples were collected, washed with saline, and then placed in clean, sterile test tubes for further liver function tests. Duodenal segments were obtained and the intestinal digest was immediately removed and stored at −20 °C for enzymatic assays. For the gut microbiota analysis, cecum segments were rapidly excised and the contents (1 to 1.5 g per bird) were collected in 2 sterilized centrifuge tubes (1.5 mL). The excised cecum segments were then cut and gently rinsed with sterilized saline (0.75%), and mucosa (0.6 to 0.8 g per bird) was collected in 2 sterilized centrifuge tubes (1.5 mL) by scraping the jejunal segments with sterilized forceps. The contents of the cecum and mucosa were snap-frozen in liquid nitrogen and then stored at −80 °C for sequencing.

### 2.7. Measurement of Physiological Indicators

Precollected liver tissues were homogenized with 4 volumes of phosphate buffer (PBS) and centrifuged at 1000× *g* for 5 min. The supernatant was used for malondialdehyde (MDA, Suzhou Grace Biotechnology Co., Ltd., Suzhou, China) content, catalase (CAT, Suzhou Grace Biotechnology Co., Ltd., China) activity, superoxide dismutase (SOD, Suzhou Grace Biotechnology Co., Ltd., China) activity, total antioxidant (T-AOC, Suzhou Grace Biotechnology Co., Ltd., China) activity, and glutathione reductase (GSH-Px, Suzhou Grace Biotechnology Co., Ltd., China) activity evaluations by using commercial kits. The stored serum was used to measure immunoglobulin G (IgG), interleukin-6 (IL-6), interleukin-1β (IL-1β), and tumor necrosis factor α (TNF-α) levels by using commercial kits (Quanzhou Jiubang Biotechnology Co., Ltd., Quanzhou, China). Digestive fluid was collected from the duodenal segments, and changes in amylase and lipase levels induced by supplementation with *L. paracasei* XLK401 were determined using an ELISA micro α-amylase activity assay kit (Quanzhou Jiubang Biotechnology Co., Ltd., China) and lipase (LPS trace method) activity assay kit (Quanzhou Jiubang Biotechnology Co., Ltd., China).

### 2.8. DNA Extraction and 16S rRNA Gene Sequencing Analysis

A total of six samples of chicken cecal contents were collected for the microbiome analysis, as described above, for the sample collection. Microbial total genomic DNA was extracted from the samples using the MagMAX™-96 DNA test kit (Thermo Fisher Scientific, Shanghai, China), according to the manufacturer’s instructions. The 16S rRNA V3–V4 region was amplified and sequenced at Majorbio Bio-Pharm Technology Co., Ltd. (Shanghai, China) on the Genomic Analysis Platform-IBIS using Illumina MiSeq paired-end technology. Sequences were analyzed in the Ubuntu terminal using the UPARSE method [19], merging raw reads. Moreover, operational taxonomic units (OTUs) were clustered using UPARSE based on a similarity threshold greater than 97%. The RDP classifier algorithm was used to compare OTU representative sequences with 97% similarity against the SILVA (Version 138, Shanghai, China) database for the taxonomic analysis. The analyses related to the 16S data (i.e., diversity, phylum level, genus level, correlation analysis, and PICRUSt2 analysis) were performed on the Megabio online analysis platform (Shanghai, China).

### 2.9. Statistical Analysis

The data were statistically analyzed using GraphPad Prism 8.0 (GraphPad software). Values are expressed as the mean ± standard deviation (SD). The independent samples *t*-test (two-tailed test) was used to assess differences between groups. In all statistical tests, differences with *p*-values less than 0.05 were considered significant: * *p* < 0.05, ** *p* < 0.01, *** *p* < 0.001. Associations between biochemical index data and the relative abundance of cecal microbial genera were examined by correlation heatmap analysis.

## 3. Results

### 3.1. Draft Genome Sequencing, Assembly, and Mapping

Whole genome sequencing showed that the genome of *L. paracasei* XLK401 consisted of a single circular chromosome (3101562 bp in total) with a GC content of 46.35% (Appendix A). The gene analysis of *L. paracasei* XLK401 using the KEGG database (Appendix A), with the highest content in the ‘metabolism’ category (70.1%), followed by ‘environmental information processing’ (15.5%), ‘genetic information processing’ (12.0%), ‘organismal systems’ (1.5%), and ‘cellular processes’ (0.9%). In addition, the COG (protein orthologous grouping) analysis revealed (Appendix A) that a total of 2941 protein-coding genes were classified into 21 categories, which were mainly involved in carbohydrate transport and metabolism (218 genes), amino acid transport and metabolism (141 genes), and translation, ribosome structure, and biogenesis (141 genes). Analysis using both the KEGG and COG databases indicated that the most abundant genes in the *L. paracasei* XLK401 genome were those related to metabolism, with genes associated with carbohydrate transport and metabolism being the most copious within the metabolic genes.

### 3.2. Analysis of Carbohydrate Transport and Metabolism Genes

Protein sequences were annotated based on the CAZy database using Hmmer (Appendix A). Eighty-two genes encoding carbohydrases were successfully annotated, including 42 glycoside hydrolases, 27 glycosyltransferases, 8 carbohydrate esterases, 2 auxiliary activities, 2 polysaccharide cleavage enzymes, and 1 carbohydrate-binding module. Further analysis revealed that *L. paracasei* XLK401 had eight *gh13* subfamilies encoding sugar hydrolases that cleaved oligosaccharides or polysaccharides linked by A-1,4-glycosidic bonds. The subfamilies *GH1* and *GH2* could encode cleavage enzymes that catalyzed the formation of 1,4-glycosidic bonds; in addition, four *gh25* genomes encoding lysozyme were identified. Ten *GT2* and seven *GT4* subfamilies encoded glycosyltransferases. A carbohydrate binding module, the cbm34 subfamily, was also found.

### 3.3. Analysis of Antimicrobial Compound Genes

Three gene clusters associated with antimicrobial substance synthesis were predicted in the *L. paracasei* XLK401 genome (Figure 1). The antiSMASH database predicted the presence of one gene cluster in the genome associated with ribosomal synthesis and the post-translational modification of peptide products (*RiPP-like*). Based on the BAGEL4 platform, it was shown that the genome contained two clusters of bacteriocin synthetized genes. As core genes, *LSEI 2386*, *Enterocin X chain beta*, and *Carnocin CP52* contain five genes associated with immune genes and transport, as well as a transport and leader cleavage gene. Another cluster had *Thermophilin A* as the core gene.

### 3.4. Genomic Characterization of Probiotic Traits

Genes for the following probiotic features were examined: tolerance to stress conditions, aid in adhesion, antioxidative stress immunity, and protective repair of DNA and proteins. The genomic analysis detected 14 gene-encoding proteins that may be related to the tolerance of bile salts and acidic environments. Furthermore, the genes related to immune responses against oxidative stress, and protein and DNA molecular repair protection were also present in the genome (Table 2).

### 3.5. In Vitro Simulated Gastrointestinal Experiment

The tolerance of *L. paracasei* XLK401 in simulated gastric fluid at pH 2.0, 3.0, and 4.0, and in simulated intestinal fluid at pH 6.8 and at different concentrations of derived bile salts are shown in Table 3. The survival rate of this strain in simulated stomach juices gradually decreases with the decreasing pH, but reaches a high of 70.49% at the lowest simulated pH conditions. In addition, the survival rate of *L. paracasei* XLK401 gradually decreased with increasing bile salt concentration, but still retained a survival rate of 76.10% at the highest simulated bile salt concentration. These results suggest that *L. paracasei* XLK401 is able to cope with gastrointestinal stress and preserve high survival rates.

### 3.6. Effect of Supplementation with L. paracasei XLK401 on the Growth and Digestive Enzymes in Chickens

As shown in Figure 2A, the FGL group had a heavier body weight than the NC group in the growth performance comparison, whereas FCR did not observe a difference between the groups. Compared with the NC group, the FGL group showed a significant increase (*p* < 0.05) in α-amylase activity but showed a decrease in lipase activity (Figure 2B). These results indicate that chicks in the FGL group have a higher utilization of starch in feed and lower digestion of lipids.

### 3.7. Effect of L. Paracasei XLK401 on Liver Antioxidant Capacity and Serum Immune Factors

Figure 3 demonstrates the effect of *L. paracasei* XLK401 supplementation on the tissue antioxidant capacity of chicken liver. In the GSH-Px, SOD, and T-AOC activity analysis, the level of the FGL group was significantly higher than that of the NC group (*p* < 0.05). In the CAT activity analysis, the FGL group had lower levels than the NC group. In addition, in the analysis of MDA activity levels, the FGL group was significantly lower than the NC group (*p* < 0.05).

The level content of immune factors in the blood of chickens was examined. The results show that while the supplementation with *L. paracasei* XLK401 does not significantly change the level contents of IL-1β, IL-10, and TNF-α in chickens compared to NC treatment, *L. paracasei* XLK401 significantly reduces the level of pro-inflammatory factor IL-6 in chicken serum (Figure 3H).

### 3.8. Effects of L. paracasei XLK401 on Chicken Gut Microbiota

Alpha diversity analysis of chicken cecum showed that the Ace and Chao indices, which represent the number of OTUs in the community, were slightly higher in the NC group than in the FGL group, and the Shannon and Simpson indices, representing the diversity of the colony, of the FGL group were slightly higher than those of the NC group. However, none of the differences were significant; therefore, there was no significant alteration in the diversity or abundance between the two groups (Appendix A). Principal component (PCoA) in β-diversity analyses showed that the biomes of the NC and FGL groups were divided into two different fractions and therefore differed between the two groups (Appendix A). At the phylum level, Firmicutes, Actinobacteriota, Bacteroidetes, and Proteobacteria were the most abundant (Figure 4). The FGL and NC groups did not differ significantly in the abundances of these four phyla in most phylum comparisons; however, the abundance of Actinobacteriota in the NC group was significantly higher than that in the FGL group (*p* < 0.05).

At the genus level, we mapped the top 10 dominant bacterial genera (Figure 5). Among them, *unclassified Lachnospiraceae* sp. was the most dominant bacterial genus, accounting for 7.08–10.71% of the total genera; however, the difference in the abundance of this genus between the NC and FGL groups was not significant. In the comparison of the remaining genus abundances, most were not significantly different between the two groups. However, the NC group had a higher abundance of *Bifidobacterium* sp. than the FGL group (*p* < 0.001), and the FGL group had a higher abundance of *Subdoligranulum* sp. than the NC group (*p* < 0.05).

### 3.9. Correlation of Physiological Indicators with Cecal Microbiota

To explore the relationship between gut microbes and host digestive system characteristics, antioxidant capacity, and immune status, we performed correlation analyses to assess the association between the fecal microbiota (the 10 genera with the highest relative abundance) and host physiological responses in chickens (Figure 6). In the correlation analysis, *unclassified_f__Lachnospiraceae* sp. showed a significant negative correlation with SOD and a significant positive correlation with lipase. *Ruminococcus_torques_group* sp. showed a significant positive correlation with lipase. *Blautia* sp. showed a significant negative correlation with CAT. *Bacteroides* sp. showed a significant negative correlation with GSH-Px and α-amylase, and a significant positive correlation with IL-10. *Collinsella* sp. was significantly and positively correlated with lipase. *Bifidobacterium* sp. showed a significant negative correlation with GSH-Px and α-amylase, and a significant positive correlation with IL-6. *Enterococcus* sp. was significantly and positively correlated with IL-10. *Subdoligranulum* sp. showed a significant negative correlation with IL-1β and MDA, and a significant positive correlation with SOD. *Barnesiella* sp. was significantly and positively correlated with SOD.

### 3.10. Functional Prediction of Bacterial Communities

Organic life responses are always accompanied by a variety of metabolic pathways. To investigate the impact of gut microbe metabolic reactions on the host, we used PICRUSt2 to predict the bacterial function of chick gut microbes based on the KEGG database. Among them, based on the bacterial sequence’s relative abundance in the samples, most of the predicted pathways could be divided into six functional groups (pathway level 1): metabolism (76.86%–76.57%), genetic information processing (8.55%–8.46%), environmental information processing (76.86%–76.57%), processing (6.21%–5.75%), cellular processes (4%–3.89%), human diseases (3.18%–3.10%), and organismal systems (1.71%–1.68%) (Figure 7A). Level 2 KEGG direct homology functions were predicted to include 11 metabolisms, 2 human diseases, 3 genetic information processing, 2 environmental information processing, 3 cellular processes pathways, and 1 organismal system with a relative abundance above 1% (Figure 7B). Among the major metabolic pathways were carbohydrate metabolism (17.21%–17.24%), amino acid metabolism (11.70%–12.07%), energy metabolism (6.85%–6.94%), metabolism of cofactors (6.62%–6.75%), and vitamins and membrane transport (6.01%–6.55%). These results suggest that microorganism metabolic pathways in the chicken gut are more frequently expressed in metabolic pathways associated with carbohydrate metabolism.

## 4. Discussion

Using probiotics as growth promoters in place of antibiotics in animal feed has increased dramatically over the years [20]. However, the properties of probiotics have been proven to be strain specific, making it crucial to assess new strains for specific beneficial properties [21]. In this study, the genomic characterization of the potential probiotic *L. paracasei* XLK401 was evaluated and its in vivo probiotic potential was assessed using a chicken feeding experiment. The results show that strain XLK401 is indeed an excellent feed additive.

Firstly, when choosing potential probiotic strains for viability and functionality in the gut, the ability to tolerate acid and bile salt is a crucial factor [22]. While passing through the stomach, these probiotics must withstand acidic levels as low as pH 3 to reach the lower GI tract and should remain effective for a minimum of 4 h [23]. The probiotic should also remain viable in the presence of 0.3% bile salts [24]. Therefore, strain XLK401, possessing multiple genes related to bile salt and acid tolerance (*Asp23*, *atpD*, *atpA*, *atpH*, and *atpF*), demonstrated strong probiotic properties under acidic (pH 2.0) as well as 0.3% bile salt conditions. In addition, the adaptation to bile salts was shown to be associated with carbohydrases and glycosidases [25]. The presence of carbohydrate metabolism-related genes in probiotics is also important for gut microbial–host interactions, which enhance microbial survival [26]. Glycoside hydrolases play a key role in the hydrolysis of carbohydrate glycosidic bonds, while glycosyltransferases are involved in the biosynthesis of disaccharides, oligosaccharides, and polysaccharides that contribute to glycosidic bond formation [27]. In this study, the genome of *L. paracasei* XLK401 contained a large number of carbohydrate-related genes, as well as genes encoding glycosidases, such as *GH* and *GT*, which enabled the strain to synthesize carbohydrates and metabolize sugars.

Another important characteristic of potential probiotics is antimicrobial activity, where probiotics can maintain intestinal homeostasis by inhibiting the growth of intestinal pathogenic bacteria [28] This study’s AntiSMASH 5.0 and BAGEL 4.0 predictions revealed three gene clusters linked to the production of antimicrobial compounds in the *L. paracasei* XLK401 genome. These contained ribosomes (*RiPPs*), *Enterocin X chain beta*, *Carnocin CP52*, and *Thermophilin A*. *Enterocin X chain beta*, which belonged to the enterococci group, which are ribosomal peptides that are thought to kill or inhibit other microorganisms’ growth [29]. *Thermophilin A* is a relatively heat-stable and significantly glycosylated bacteriocin with a bactericidal mode of action against sensitive cells [30]. Thus, these results suggest that *L. paracasei* XLK401 has the potential to antagonize bacteria and can effectively influence the balance of intestinal flora.

This study investigated the in vivo effects of *L. paracasei* XLK401 on chicks by supplementing their diets with *L. paracasei* XLK401. There was a significant improvement in the body weight of broilers in the treatment group as compared to the blank control group. It is possible that probiotics increase digestive enzyme activities, such as amylase, and secrete some unknown growth-promoting factors that promote animal growth [31]. In the results of this experiment, it was demonstrated that the amylase activity of the treated group was significantly higher than that of the control group. In addition, it was shown that some *L. paracasei* strains could improve the immune function of animals [32]. Cytokines are thought to play an important role in cell-mediated immune responses. Helper T-cell factors, including IL-2, TNF-α, and IL-6, drive cell-mediated immune responses [33]. In our study, IL-1β, TNF-α, and IgG levels were reduced and IL-6 concentrations were significantly lower in the FGL group compared to NC, whereas the high production of pro-inflammatory cytokines (IL-6) was an indicator of inflammation [34]. This suggests that the administration of *L. paracasei* XLK401 reduces the level of inflammation and maintains immune homeostasis.

Previous research has shown that oxidative stress can compromise meat quality and fertility, posing serious threats to livestock production [35]. Therefore, reducing oxidative stress is essential to ensure poultry health. Probiotics have been widely demonstrated to enhance the antioxidant capacity of the host by increasing the activity of antioxidant enzymes or activating antioxidant-related pathway enhancements of the host antioxidant capacity [36]. In this study, feeding diets containing *L. paracasei* XLK401 significantly increased antioxidant enzyme (SOD and GSH-Px) and T-AOC activities, as well as significantly reduced chicken MDA levels. MDA is an important product of lipid peroxidation, and the level of MDA is a measure of the extent of oxidative damage [37]. In addition, based on the analysis of the sequencing data, strain XLK401 was found to contain a variety of antioxidant-related genes, including the NADPH-dependent oxidoreductase genes trxA and trxB, and the Thiol peroxidase gene, tpx, while the study confirmed that the expression of these gene (ahpC, ahpF, bcp, trxB, trxA, nrdH, and msrAB) products were the active ingredients of the antioxidant system [38].

The gut microbiota is a critical part of the gastrointestinal tract and contributes significantly to host health [39]. Probiotic supplementation is gaining a significant amount of attention due to its ability to alter gut microbiota and consequently improve chicken health [40]. In the present study, no significant effect of *L. paracasei* XLK401 on the α-diversity of chicken cecum flora was observed; however, the abundance of cecum flora was significantly affected. In the FGL group, Actinobacteria abundance was significantly decreased, *Bifidobacterium* sp. abundance was also significantly decreased and *Subdoligranulum* sp. abundance was significantly increased, whereas probiotics could modulate the abundance of certain bacteria without altering the overall microbial structure. On the other hand, dietary supplementation with *Bacillus amyloliquefaciens* US573 decreased the abundance of Actinobacteria in the gut of European sea bass [41]. Feeding *Lactobacillus royceae* KUB-AC5 to growing broilers increased the abundance of Actinobacteria in the gut [42]. On the other hand, *Lactobacillus plantarum* CCFM8724 altered the oral microbiota and decreased the abundance of *Bifidobacteria* in children with dental caries [43]. *Lactobacillus plantarum* HY7714 intake reduces the abundance of *Bifidobacterium* bifidum in the gut [44]. It has been suggested that *Subdoligranulum* sp. may be the best probiotic strain because it is a spore-forming butyric acid producer [45]. *Enterococcus* sp. and *Subdoligranulum* sp. are beneficial to ameliorating necrotizing enterocolitis (NEC) by affecting bacteriophage and butyrate production, respectively [46]. It was found that butyrate had an anti-inflammatory potential [47]. In the correlation analysis of this study, the content of *Subdoligranulum* sp. was negatively correlated with the inflammatory factors IL-1β and MDA and positively correlated with SOD, which has an anti-inflammatory capacity. These results suggest that supplementation with *L. paracasei* XLK401 can modulate the microbial community, while some microorganisms have the potential to alleviate intestinal inflammation.

The functional prediction results show that the relative abundance of “carbohydrate metabolism”, which belong to metabolism, is significantly higher in the intestine. Carbohydrate metabolism genes are related to feed efficiency, while feed utilization efficiency is linked to chicken growth, and good feed utilization efficiency can promote chicken growth [48]. Importantly, the functional pathways were predicted from 16S data and further functional validation studies should be conducted in the future. This study was based on the genome-wide data from *L. paracasei* XLK401 and bioinformatics analysis of the gut microbiome of chicks, and elucidated the mechanisms by which *L. paracasei* XLK401 promoted host health. However, the study still had some drawbacks, such as small sample sizes and incomplete health mechanisms. These limitations necessitate the expansion of the sample size to understand the metabolic pathways and functional changes in probiotics using metabolomics and transcriptomics to provide directions for further research on the production and utilization of probiotics.

## 5. Conclusions

In summary, this study was the first to characterize the genomic properties of XLK401. We found that the XLK401 genome housed genes related to acid resistance and antioxidants. In addition, we performed the functional prediction of the genome and found that some genes were involved in the synthesis of carbohydrates and active enzymes for metabolizing sugars, as well as clusters of genes related to antimicrobial compounds. Strain XLK40 improved the level of inflammation in chicks, maintained immune homeostasis, as well as increased antioxidant potential, which may be related to the gene expression of strain XLK401. However, further validation is needed. Feeding strain XLK401 also modulated the chick gut microbiota, Actinobacteria abundance was significantly decreased, *Bifidobacterium* sp. abundance was also significantly decreased, and *Subdoligranulum* sp. abundance was significantly increased. *Subdoligranulum* sp. has the potential to reduce intestinal inflammation. These findings, which demonstrate that *L. paracasei* XLK401 has probiotic potential to enhance growth performance in chickens, are a potential strategy to improve animal growth performance.

## Figures and Tables

**Figure 1 microorganisms-11-02140-f001:**
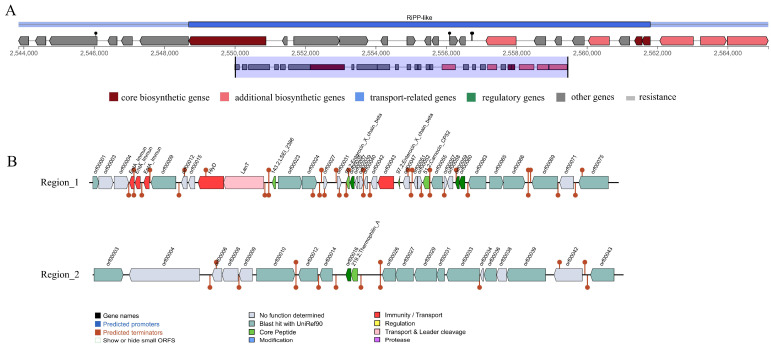
Prediction of antimicrobial-associated protein structures in the genome of *L. paracasei* XLK401. (**A**) Synthetic gene cluster of ribosomal synthesis and post-translational modification of peptide products (*RiPP-like*; based on antiSMASH database prediction). (**B**) Two bacteriocin (region_1: *LSEI 2386*, *Enterocin X chain beta*, *Carnocin CP52*, and region_2: *Thermophilin A*) synthesis gene clusters (predicted based on the BAGEL4 database).

**Figure 2 microorganisms-11-02140-f002:**
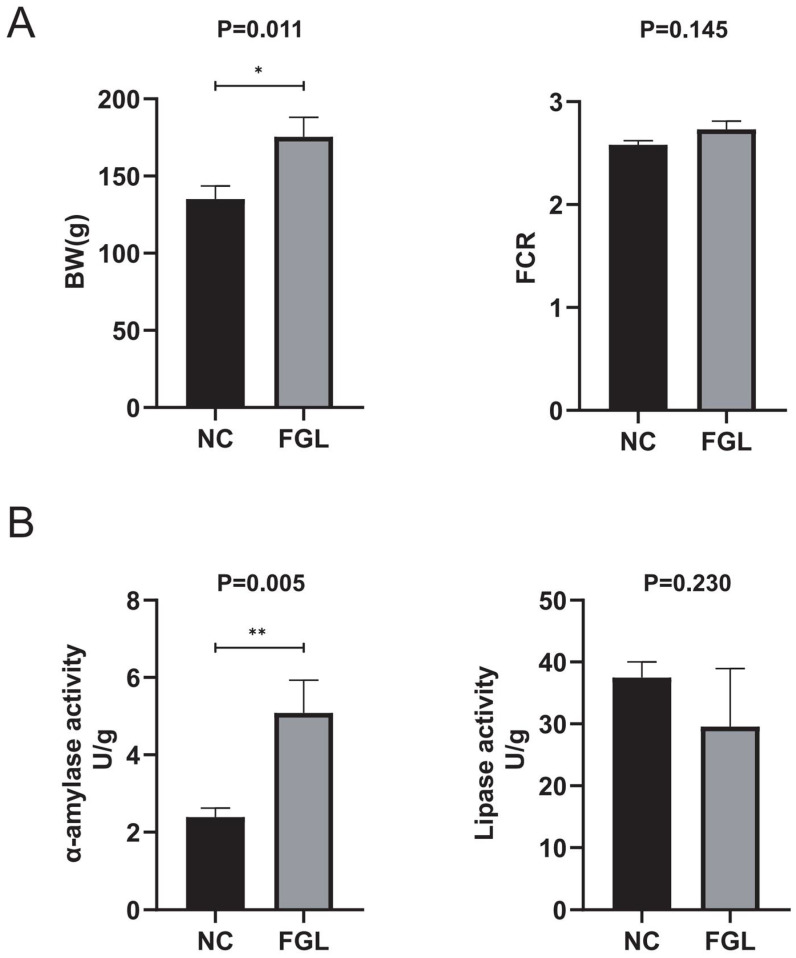
Effect of feeding diets containing *L. paracasei* XLK401 on chick growth and digestive enzymes. Final body weight (BW) results and feed conversion ratio (FCR = feed consumption/weight gain) of chicks are indicated in (**A**). α-amylase and lipase activities are indicated in (**B**). * *p* < 0.05, ** *p* < 0.01.

**Figure 3 microorganisms-11-02140-f003:**
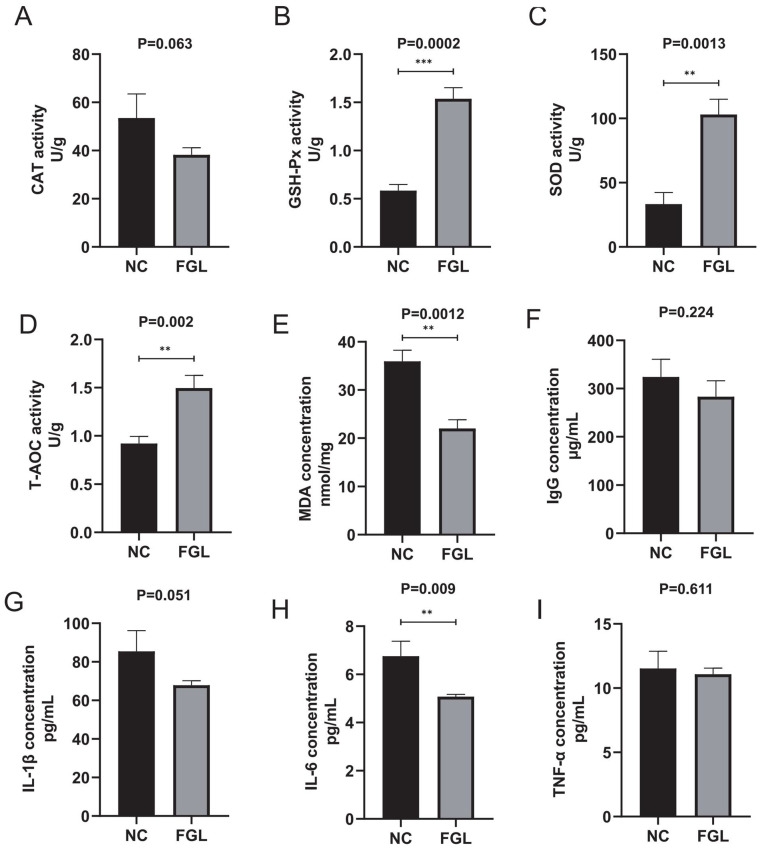
Effect of *L. paracasei* XLK401 on antioxidant enzyme activity and MDA levels in chickens, and the induction of serum immune factors. (**A**) CAT activity. (**B**) GSH-Px activity. (**C**) SOD activity. (**D**) T-AOC activity. (**E**) MDA level content. (**F**) IgG concentration. (**G**) IL-1β concentration. (**H**) IL-6 concentration. (**I**) TNF-α concentration. ** *p* < 0.01, *** *p* < 0.001.

**Figure 4 microorganisms-11-02140-f004:**
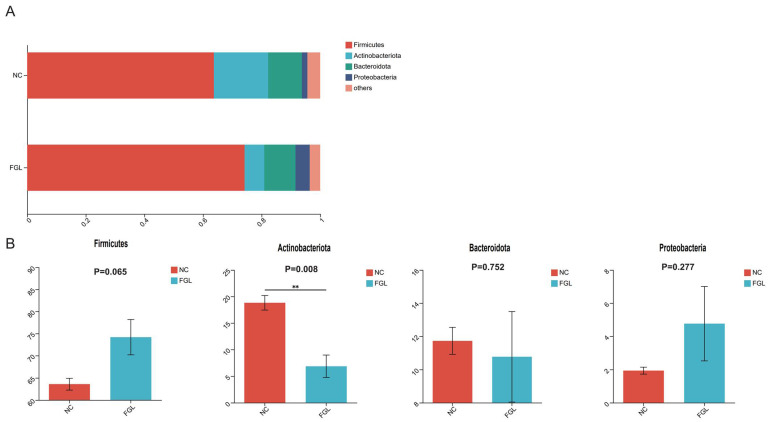
Composition of the phylum of the gut microbiota in chickens. (**A**) Relative abundance of the first five phyla. (**B**) Comparative differences at the phylum level of the gut microbiota. ** *p* < 0.01.

**Figure 5 microorganisms-11-02140-f005:**
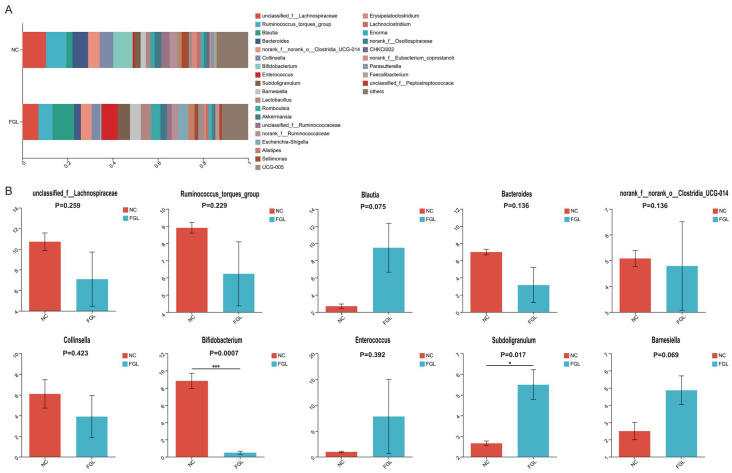
Composition of the genus of the gut microbiota in chickens. (**A**) Relative abundance of the first 28 genus. (**B**) Comparative genus-level differences in the top 10 gut microbiota. * *p* < 0.05, *** *p* < 0.001.

**Figure 6 microorganisms-11-02140-f006:**
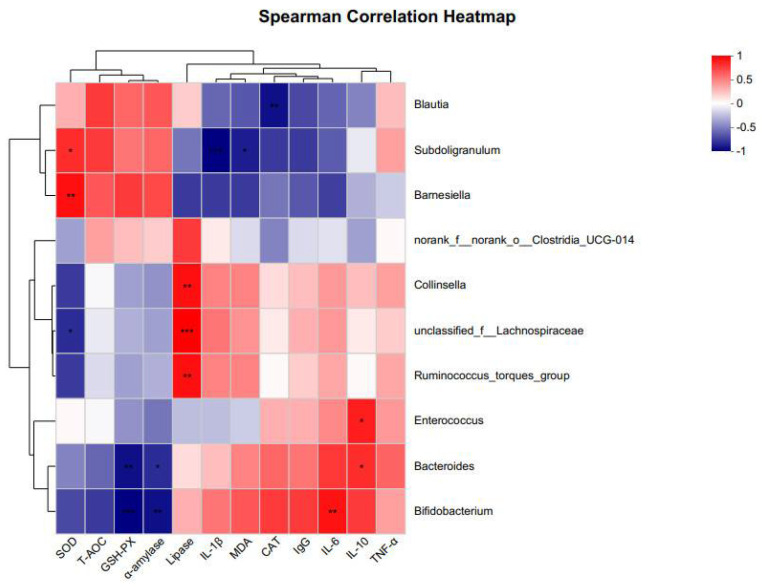
Heat map of correlation coefficients between gut microbiota and digestive characteristics, antioxidant enzyme activity, and immune factors of chickens at 21 days. “red” indicates a positive correlation (*p* < 0.05) and “blue” indicates a negative correlation (*p* < 0.05) * *p* < 0.05, ** *p* < 0.01, *** *p* < 0.001.

**Figure 7 microorganisms-11-02140-f007:**
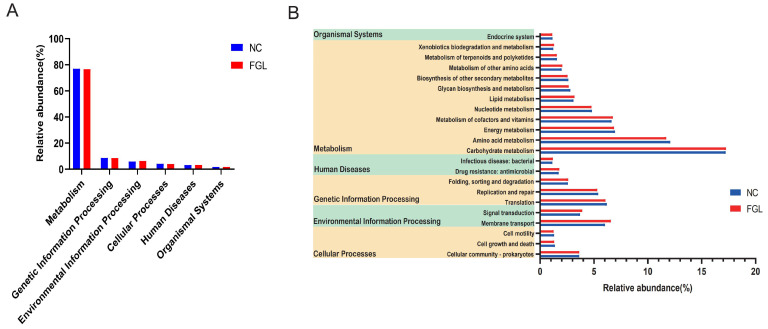
Functional prediction of bacteria in the chicken gut annotated by PICRUSt2. (**A**) Primary functional class; (**B**) secondary functional class (relative abundance >1%).

**Table 1 microorganisms-11-02140-t001:** Ingredient composition and nutrient contents of basal diets.

Ingredients, %	1–21 d
Corn	55.41
Soybean meal	31.50
Palm oil	5.00
Phosphorus	3.60
Calcium	1.30
Salt	0.34
Lysine HCL	1.40
Methionine	0.22
Arginine	0.03
Vitamin–mineral premix	0.50
Limestone	0.60
Sodium carbonate	0.10
Metabolizable energy (MJ·kg^−1^)	14.01
Crude protein	20.0
Calcium	1.00
Total phosphorus	0.55
Lysine total	1.41
Methionine	0.50

Supplied per kilogram of diet: vitamin A (trans-retinyl acetate), 10,050 IU; vitamin D3, 2800 IU; vitamin E (DL-α-tocopheryl acetate), 50 mg; vitamin K3, 3.5 mg; thiamine, 2.5 mg; riboflavin, 7.5 mg; pantothenic acid, 15.3 mg; pyridoxine, 4.3 mg; vitamin B12 (cyanocobalamin), 0.02 mg; niacin, 35 mg; choline chloride, 1000 mg; biotin, 0.20 mg; folic acid, 1.2 mg; Mn, 100 mg; Fe, 85 mg; Zn, 60 mg; Cu, 9.6 mg; I, 0.30 mg; Co, 0.20 mg; and Se, 0.20 mg.

**Table 2 microorganisms-11-02140-t002:** Probiotic-related genes present in *L. paracasei* XLK401.

Gene	Putative Function	Locus Tag
Oxidative stress resistance		
*trxA*	NADPH-dependent oxidoreductase activity	NCY29_04240
*trxB*	NADPH-dependent oxidoreductase activity	NCY29_04995
*tpx*	Thiol peroxidase	NCY29_04015
pH stress resistance		
*Asp23*	Alkaline shock protein (Asp23) family	NCY29_08625
*atpD*	F0F1 ATP synthase subunit beta	NCY29_06455
*atpA*	F0F1 ATP synthase subunit alpha	NCY29_06445
*atpH*	F0F1 ATP synthase subunit delta	NCY29_06440
*atpF*	F0F1 ATP synthase subunit B	NCY29_06435
DNA and protein protection and repair		
*clpb*	Persistence capacity in vivo	NCY29_07365
*clpP*	Persistence capacity in vivo	NCY29_05065
*msrB*	Persistence capacity in vivo	NCY29_08150
*rfbB*	Low pH tolerance	NCY29_10470
Adhesion ability		
*dnaK*	Mucin binding	NCY29_08360
*gndA*	Promotes adherence to epithelial cells	NCY29_08930

**Table 3 microorganisms-11-02140-t003:** Tolerance of *L. paracasei* XLK401 to simulated gastrointestinal fluids and different bile salt concentrations.

Treatment	Time(h)	Survival Rate
Gastric juice (pH 2)	5	70.49 ± 2.42%
Gastric juice (pH 3)	5	82.95 ± 2.00%
Gastric juice (pH 4)	5	89.21 ± 0.24%
0.3% Bile salt	4	94.75 ± 0.62%
0.6% Bile salt	4	87.46 ± 0.46%
0.9% Bile salt	4	76.10 ± 1.62%

## Data Availability

The datasets presented in this study can be found in online repositories. The names of the repository/repositories and accession number(s) can be found at: https://www.ncbi.nlm.nih.gov/, uploaded 6 March 2023, CP098411.1, SAMN335764(17-22), SRR24102167(5-7).

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
