# Peer review of "Genome-Wide and 16S rRNA Sequencing-Based Analysis on the Health Effects of Lacticaseibacillus paracasei XLK401 on Chicks"

_microorganisms, 2023, doi:10.3390/microorganisms11092140_

Round 1
Reviewer 1 Report
The manuscript describes the effect of orally administration of Lacticaseibacillus paracasei XLK401 on chick growth, total antioxidant activity and antioxidants enzyme activity in liver and amilase and lipase activity in duodenum fluid. Also, the work shows the cytokine amount in serum. Moreover, Lacticaseibacillus paracasei XLK401 genome was sequenced and the relative amount of the microorganism present in the chick control and treated cecal samples was analyzed. However, it would be interesting that the authors improve the discussion. The work should explain if there is a correlation between the genes present in the L. paracasei strain and the enhanced antioxidant status of the treated chicks. Furthermore, although the genes for the synthesis of bacteriocins were found in the genome there is no information if these kind of antimicrobial compounds are active against the microbial genera that are in lower amount in the treated group. On the other hand, I think section 2.8. (Correlation of physiological indicators with cecum microbiota) is too speculative. Is there really a correlation between the activity of the antioxidant enzymes in the host with the relative abundance of the different microbial genera? Why? Did the authors analyzed the genomes of the different genera in silico, using database? Which one?
Minor comments:
Results:
-Line 81: delete “the whole”
-Line 136-139: please rewrite the sentence is confusing, I think you wanted to say that the BW was higher in the FGL group respect to the NC group, while no differences were observed for FCR between groups.
-Lines 136-142: please add the meaning for FGL, NC and BW, FCR because is the first time you use the abbreviations.
-Add a comment about lipase activity.
-Add a comment about catalase activity
-Line 187: replace “highly significant” by showed “a higher abundance than”
-Line 189: replace “was significant” by “was higher”
-Line 201: delete “also”
-Line 203: how are Bacteroides and Bifidobacterium asociated with alpha- amilase? How dose the abundance of this bacteria genera affect the expression or activity of this enzime in the host? Add references or how did you analyze the results (genomic sequences for these genera?).
-Line 320: how do you know that L. paracasei XLK401 inhibited the expression of antiobiotic resistance genes in Bifidobacterium?
-Line 465: replace “is derived” by “could be derived by the expression” because no proteomics or transcriptomics studies were done in this work.
Reviewer 2 Report
1. Lines 346-350 Rewrite these lines. Describe in detail how the L. paracasei was isolated.
2. Lines 352-370. Describe in detail and in an orderly manner how the WGS assembly and annotations were performed. You can consult already published papers to double-check how to improve your writing and the choice of words to use.
3. Lines 372-383. Sounds confusing, rewrite by looking at similar articles online.
4. Lines 372-383. Change (2) Positive control to FGL group. You have just a Negative control; that is without L. paracasei and a treatment group with L. paracasei. The inclusion of L. paracasei makes it your treatment group.
5. Lines 404-413. rewrite this part by improving on the language. delete fingers in line 410.
6. Lines 446-455. rewrite the statistical analysis section, and report on only the statistical analysis that you used in this study.
7. lines 457-476. Your conclusion must be rewritten.
8. Lines 318-322. You will need more references to support your conclusion that Bifidobacterium is a pathogen in chicks. There are several species or strains of Bifibacterium most of which are proven to have probiotics properties in human studies. Get more literature to support your findings.
9. Same in the case of Actinobacteria. Its pathogenicity. I will advise you report on what you observed, which is a negative correlation, and we know for sure that correlation is not causality.
The results and discussion section are well written and does not need much language editing but I advise you to work on the Materials and Method section to improve both the language and the choice of wording.
Round 2
Reviewer 1 Report
The manuscript has been corrected taking into account the reviewers recommendations. However, some minimal corrections have to be done for approval
Lines 28-29: Among them, Actinobacteria abundance in chicken intestine after feeding with L. paracasei XLK401 was significantly decreased. Bifidobacterium abundance was also
Line 30-31: delete this sentence it was written twice.
Line 49-50: correct verb tense
Introduction:
Please, once the species was defined abbreviate to L. paracasei in the rest of the manuscript sections
Materials and methods:
Lines 107-103: delete this paragraph the strain isolation was not done in this work and is not relevant for the current manuscript.
Line 166: delete one ) from (0.3, 0.6, and 0.9% (w/v))
Line 194: delete one ) from (1×107 CFU/g feed)) (10/cage).
Lines 216, 239 replace rpm by x g
Line 254: Rephrase the sentence
Line 367: Capital letter for Final
Lines 388-389, 390: what do you mean with “representing the abundance of the colony” do you mean the abundance of the species or bacterium genus? Or the abundance of the bacteria species in the colon? Please rephrase
Line 544 replace “Lacticaseibacillus paracasei “ by some L. paracasei strains
Line 547 delete “that”
Line 554: delete “And”
line 563: capital letters for “While”
Lines 565-566: delete this sentence
Line 623: delete “For example”
Line 624: delete parenthesis from “(Bacillus amyloliquefaciens US573)”
Line 625: lowercase letter for “Royceae”
Add sp. to Subdoligranulum, Bifidobacterium and Enterococcus when the species is not determined or when you are writting about aspects related to the bacterial genera
English language has been improved, but needs some further editing
